# A Dansyl-Modified Sphingosine Kinase Inhibitor DPF-543 Enhanced De Novo Ceramide Generation

**DOI:** 10.3390/ijms22179190

**Published:** 2021-08-25

**Authors:** Maftuna Shamshiddinova, Shokhid Gulyamov, Hee-Jung Kim, Seo-Hyeon Jung, Dong-Jae Baek, Yong-Moon Lee

**Affiliations:** 1College of Pharmacy, Chungbuk National University, Chungbuk 28160, Korea; shamshiddinovamaftuna@gmail.com (M.S.); shohid9395@gmail.com (S.G.); rlagmlwnd94@naver.com (H.-J.K.); sherry987@naver.com (S.-H.J.); 2College of Pharmacy, Mokpo National University, Jeonnam 58628, Korea; dbaek@mokpo.ac.kr

**Keywords:** PF-543, ceramide, SPHK, SPT, ceramide synthases, sphingolipid, metabolism, LC-MS/MS

## Abstract

Sphingosine-1-phosphate (S1P) synthesized by sphingosine kinase (SPHK) is a signaling molecule, involved in cell proliferation, growth, differentiation, and survival. Indeed, a sharp increase of S1P is linked to a pathological outcome with inflammation, cancer metastasis, or angiogenesis, etc. In this regard, SPHK/S1P axis regulation has been a specific issue in the anticancer strategy to turn accumulated sphingosine (SPN) into cytotoxic ceramides (Cers). For these purposes, there have been numerous chemicals synthesized for SPHK inhibition. In this study, we investigated the comparative efficiency of dansylated PF-543 (DPF-543) on the Cers synthesis along with PF-543. DPF-543 deserved attention in strong cytotoxicity, due to the cytotoxic Cers accumulation by ceramide synthase (CerSs). DPF-543 exhibited dual actions on Cers synthesis by enhancing serine palmitoyltransferase (SPT) activity, and by inhibiting SPHKs, which eventually induced an unusual environment with a high amount of 3-ketosphinganine and sphinganine (SPA). SPA in turn was consumed to synthesize Cers via de novo pathway. Interestingly, PF-543 increased only the SPN level, but not for SPA. In addition, DPF-543 mildly activates acid sphingomyelinase (aSMase), which contributes a partial increase in Cers. Collectively, a dansyl-modified DPF-543 relatively enhanced Cers accumulation via de novo pathway which was not observed in PF-543. Our results demonstrated that the structural modification on SPHK inhibitors is still an attractive anticancer strategy by regulating sphingolipid metabolism.

## 1. Introduction

Sphingolipid metabolism initiates via the de novo synthesis pathway at the cytosolic leaflet of the ER by serine palmitoyltransferase (SPT), which catalyzes L-serine with palmitoyl-CoA in the presence of pyridoxal phosphate and forms 3-ketodihydrosphingosine (KDS) [1]. Alternately, KDS reductase reduces the ketone group of KDS into hydroxyl group in an NADPH-dependent manner and produces dihydrosphingosine (DHS) [2]. Subsequently, six distinct ceramide synthases (CerSs) turn DHS into dihydroceramides (DHCers) by acylation [3,4]. Formed DHCers vary on fatty acid chain length depending on which type of CerS catalyzed the reaction. For instance, CerS1 produces N-stearoyl-D-erythro-sphingosine, C18-ceramide (d18:1/18:0); CerS2 is responsible for long chain ceramides including C20 and C24 [5,6,7]. On the other hand, CerS3 prefers very long chain acyl CoAs, mainly C26-ceramide, and is highly expressed in the testis [8]. Unlike CerS1, CerS4 also generates ceramides with C20 and C22 and alteration of CerS4 expression is highly associated with skin diseases [9,10,11]. Eventually, CerS5 and CerS6 produce C16-ceramide, as both have privilege towards palmitoyl CoA as a substrate [12,13]. Afterwards, DHCer desaturase 1/2 (DES1/2) enzymes in endoplasmic reticulum (ER) membrane convert the dihydrosphingoid bases into ceramides [14]. Formed ceramide is transported from the ER to the Golgi by vesicular transportation or via ceramide transport protein CERT [15,16,17]. Ceramide is a central hub in sphingolipid metabolism, which is involved in formation of complex glycosphingolipids, sphingomyelin (SM), and ceramide-1-phosphate. Another catabolic pathway of ceramide, so-called salvage pathway, produces a bioactive signaling molecule sphingosine-1-phosphate (S1P) [18]. Ceramide eventually catabolizes to sphingosine by three different ceramidases, which are classified by their pH optima [19]. Sphingosine kinase (SPHK) enzymes phosphorylate the 1-hydroxyl group of sphingosines and produces S1P. There are two isoforms of SPHK found in mammals: SPHK1 is localized in cytoplasm and SPHK2 is found in nucleus [20]. SPHK/S1P axis is actively involved in cell survival, proliferation, cancer metastasis, inflammation, type II diabetes, and cardiovascular diseases [21,22]. SPHK is overexpressed in hyperproliferative diseases, and its metabolite S1P progresses cancer cell proliferation and metastasis [21]. Thus, designing potential therapeutic chemicals which regulate SPHK/S1P signaling pathway has become a trending topic among scientists. Myriad SPHK selective inhibitors were designed and applied to clinical trials. For example, fingolimod (FTY720) SPHK1 selective inhibitor is widely used as an anti-sclerotic drug which activates protein phosphatase 2A and suppresses cancer cell growth [23]. PF-543 is the most potent SPHK 1 inhibitor with a nonlipid structure designed by Pfizer Co. In spite of high inhibitory activity (IC_50_ 2.0 nM), this compound demonstrated low efficacy in certain types of cancer cell lines, possibly due to the accumulated cellular sphingoid bases [24]. PF-543 treatment lowered the SPHK1 expression in Ca9-22 and HSC-3 cells, and decreased cell proliferation in a time- and dose-dependent manner. Furthermore, long-term incubation caused the induction of autophagy and prevented necrotic cell death [25]. In vivo application of PF-543 improved the symptoms and pathological changes in dextran-sodium-sulfate-induced ulcerative colitis in murine models. Moreover, it showed an anti-inflammatory response by depleting the level of IL-1b and IL-6 [26]. PF-543 decreased the expression of profibrotic markers such as mtDNA damage and fibrogenic monocyte recruitment in mice lungs with pulmonary fibrosis induced by bleomycin and asbestos. Moreover, post-treatment of lung epithelial cells with PF-543 suppressed pulmonary fibrosis at the expense of reduced lung mtDNA damage and monocyte recruitment [27]. Furthermore, SPHK1 inhibition by PF-543 impaired YAP1 co-localization with FSP1 in mice lung fibroblasts. In vitro studies revealed that PF-543 treatment reduced the TGF-β- or BLM-induced mitochondrial reactive oxygen species (mtROS) in human lung fibroblasts (HLFs) and the expression of fibronectin (FN) and alpha-smooth muscle actin (α-SMA). These results suggest that PF-543 attenuated the TGF-β-induced YAP1 activation and mtROS generation, causing fibroblast activation, a vital inducer of pulmonary fibrosis [28]. SPHK1 inhibition by PF-543 decreased matrix mineralization, alkaline phosphatase activity, and the mRNA expression of Runx2 and Bglap in chondrocytes and osteoblasts, making it one of the promising candidates for spondyloarthritis treatment [29]. Intraperitoneal administration of PF-543 improved endothelial function of arteries of hypertensive mice by decreasing endothelial nitric oxide synthase phosphorylation. Importantly, pharmacological inhibition of SPHK1 by PF-543 also reduced cardiac hypertrophy and endothelial dysfunction, which were induced by Ang II [30]. Another study showed that PF-543, as a specific inhibitor of SphK1, could partially minimize the detrimental effects on lung injury of cecal ligation and puncture mice. PF-543 suppressed the SPHK1/S1P axis and by this mitigated the lung injury caused by sepsis in acute ethanol intoxication in rats [31].

Various inhibitors were designed based on the core structure of PF-543 to refine its pharmacological efficacy. Fluorophore-labeled PF-543 analogue BODIPY-PF-543 gave the same SPHK1 inhibition efficacy as PF-543, by showing the IC_50_ values 19.92 and 11.24 nM, respectively. Confocal microscopy results proved that BODIPY-PF-543 is mainly located in the cytosol of the cells after treatment and might be useful for cell imaging [32]. For visualizing the local distribution of PF-543, DPF-543 was synthesized by labeling PF-543 structure with 5-(dimethylamino) naphthalene-1-sulfonyl (dansyl) moiety and was recommended to also be utilized for fluorescent-based SPHK assays (Figure 1). In the SPHK1 inhibition assay, PF-543 and DPF-543 have similar IC_50_ value, 10.4 ± 3.2 nM and 12.3 ± 2.5 nM, respectively, and docking study confirmed that DPF-543 has the same binding pattern as SPHK1, likewise PF-543 [33]. Novel PF-543 derivatives were evaluated on the activity of SPHK1/2 inhibition. However, there is lack of data about their impact on other enzymes which actively take part in sphingolipid metabolism. In this report, we investigated the comparative study on the effects of local structural modification on PF-543 on the sphingolipid metabolism. In particular, we focused on the relative changes of dihydroceramides (DHCers) and ceramides (Cers) distribution because PF-543 makes a local environment on the transient SPHK substrates accumulation of cellular sphingoid bases by inhibiting SPHK1. The LC-MS/MS system was used to analyze DHCers and Cers in the LLC-PK1 cells, porcine kidney epithelial cells which have been used for sphingolipid metabolism research.

## 2. Results

### 2.1. Cell Viability Evaluation following DPF-543 Treatment

In LLC-PK1 cells, DPF-543 and PF-543 were not toxic in the tested concentration below 31.25 µM (Figure 2). The dansyl derivatization on the position far from an active site in PF-543 showed more cytotoxic effect. The cytotoxicity by DPF-543 was observed predominantly from 62.5 µM. From this concentration, the cell viability by DPF-543 began to be reduced to 74% while PF-543 treatment was still viable, showing 96% viability. The maximum gap between DPF-543 and PF-543 on the cell viability was observed when 125 µM of DPF-543 or PF-543 had been tested. Under the 500 µM or higher concentration, two chemicals gave irreversible toxicity with 5% cell viability. For further experiments on the sphingolipid metabolism and related enzyme assays, PF-543 and DPF-543 below 30.0 µM were applied to the cells.

### 2.2. DPF-543 Activates SPT In Vitro

DPF-543 and PF-543 have displayed the specific SPHK inhibition [33]. To clarify a DPF-543 cytotoxic effect, we contrived to look at the first step of sphingolipid biosynthesis initiated by SPT. The SPT activity was measured by tracing the two-deuterium (D2-) labelled sphingolipid metabolites, 3-keto sphinganine (D2) and sphinganine (D2). In these conditions, endogenous sphingolipid metabolites were also measured simultaneously. The DPF-543 treatment triggered the 3-keto sphinganine (D2) production, which was not observed by PF-543 treatment (Figure 3a). The sphinganine (D2) increase by DPF-543 reconfirmed that DPF-543 activates the SPT enzyme to synthesize new initial sphingolipid metabolites, which may produce cytotoxic sphingolipid metabolites such as ceramides. In the same condition, endogenous KDS and sphinganine (SPA) were significantly observed after DPF-543 treatment (Figure 3b). Indeed, the DPF-543 treatment showed a 4-fold increase of KDS(D2) and SPA(D2), which also provided the parallel relation to the increase of endogenous sphingolipid metabolites. Interestingly, the PF-543 treatment greatly increased the sphingosine (SPN), a metabolite from the ceramide salvage pathway, by blocking SPHK activities. Practically, PF-543 treatment showed almost 6-fold SPN accumulation from 13 pmol/mg protein to 75 pmol/mg protein.

### 2.3. DPF-543 Strongly Activates CerSs

C17-sphinganine was used as a specific substrate on CerSase activity. We exogenously spiked palmitoyl-CoA into the reaction mixture as a fatty acid resource, thus palmitoyl (C16:0) ceramide (C17 sphingoid base) was profoundly formed. Total increased amount of C17-based DHCers (C17-DHCers) were from 8.75 nmol/mg protein to 9.34 nmol/mg protein by PF-543 and 18.07 nmol/mg protein by DPF-543, relatively (Figure 4a). Next, we investigated the relative increases on individual C17-DHCers. The abundant C17-DHCers in LLC-PK1 cells were C17-DHCer with FA 16:0 (2952.7 pmol/mg protein) or 18:1 (5398.1 pmol/mg protein). Overall, PF-543 treatment did not greatly induce the C17-DHCer synthesis. Unexpectedly, the dansyl-modified DPF-543 treatment significantly raised the content of C17-DHCers 16:0 and 18:1 by 1.75-fold and 2.1-fold, respectively. Even though C17-DHCer with FA chain 18:0, 20:0, and 22:0 had a slight increase by PF-543, there was almost two-fold augmentation in other C17-DHCers (18:0, 20:0 isomer, 22:0 isomer) by DPF-543. By DPF-543 treatment, the most fold increase was observed in C17-DHCer 24:1, 24:0, and its isomers: 4-fold, 6.6-fold, and 9.5-fold, respectively. On the contrary, the content of C17-DHCer 22:0 was not altered significantly (Figure 4b).

The DHCer desaturase is the enzyme responsible for converting the increased DHCers by DPF-543 into ceramides (Cers). As the C17-DHCers were the specific substrate for DHCer desaturase, we determined the newly synthesized C17-Cers. Compared to the increased amount of C17-Cers (72 pmol/mg protein) in control groups, the PF-543 or DPF-543 treatment demonstrated higher DHCer desaturase activities on C17-Cers synthesis by the total Cer increases of 2.55-fold and 5.36-fold, respectively(Figure 4c). The Cer with fatty acid chain 16:0 and 24:0 was the major ceramide synthesized by DPF-543 treatment. Unexpectedly, the synthesis of C17-Cer 18:1 was not effectively converted from C17-DHCer 18:1, which has been a major C17-DHCer, by DPF-543 (Figure 4d). The result from DPF-543 application to the effectiveness on the new Cers synthesis was postulated as follows: that the relatively low conversion ratio by DHCer desaturase may be connected to prevent the excessive increase of cytotoxic Cers in a short period (Appendix A).

### 2.4. DPF-543 Activates aSMase

Next, we considered the activation of another Cers synthetic route by aSMase. PF-543 or DPF-543 also activated aSMase to produce fluorescent phosphocholine directly from sphingomyelin (SM) hydrolysis in this assay kit (Figure 5). The aSMase activity by DPF-543 was 1.5-fold higher than control level. The simultaneous increase of total endogenous Cers by DPF-543 was also observed (Figure 6a). The accumulation of total Cers was a sum both from de novo and from aSMase route. Of most importance, the distribution pattern of endogenous Cer analogs by DPF-543 exhibited a closed similarity on the accumulation pattern by C17-sphinganine spiking test demonstrated in Figure 4d. The endogenous Cer18:1 amount was observed to be relatively low, which was also the low synthetic rate of C17-Cer18:1 in the environment of highly accumulated C17-DHCer 18:1 by DPF-543 (Figure 6b).

### 2.5. DPF-543 Activates Cer Synthesis via De Novo Pathway

A strong SPT inhibitor, 10 μM myriocin (Myr) treatment reduced the endogenous Cers by almost 50% compared to control. However, PF-543 and DPF-543 application with Myr triggered the Cers accumulation, roughly 3.5 and 10.4 times higher than Myr itself. By comparing the data of total Cers amount by DPF-543 (Figure 6a and Figure 7a), the inhibitory contribution of Myr in Cer synthesis was not higher than expected (8.5% Cers reduced by myriocin). The relative distribution of each Cers by Myr plus DPF-543 was similar to DPF-543 single treatment (Figure 7b). The accumulated Cers by DPF-543 might be contributed via another de novo route or aSMase route.

Therefore, a strong CerS inhibitor, fumonisin B1 (FB1 35 μM), was applied to find out whether DPF-543-induced Cers accumulation is regulated or not. As shown in Figure 8a, FB1 treatment decreased almost half the amount of total Cers by DPF-543, indicating that CerSs were highly activated by DPF-543 although a part of the conversion from sphinganine (SPA) to DHCers was inhibited by FB1 treatment. The Cers distribution pattern by FB1 was almost the same as the data obtained by myriocin treatment, except for the half-reduction on Cers amount (Figure 8b). Finally, we reconfirmed the de novo steps by combined treatment of myriocin and FB1 during the DPF-543 activation on the Cer synthesis.

The total Cer amount after combined treatment was almost the same as Figure 8a, indicating the contribution of SPT activity on DPF-543-induced Cer synthesis was negligible (Figure 9a). Likewise, the Cers amount and distribution by myriocin plus FB1 treatment showed an almost similar Cers pattern obtained from FB1 treatment (Figure 9b). In our experiment, DPF-543, a PF-543 structure modified at dansylated moiety at sulphonyl position, enhanced the Cer synthesis via not only aSMase activation but also strong CerSs activation which has a responsible role for DHCers formation needed for Cer synthesis.

## 3. Discussion

SPHK1 expression and S1P formation were highly elevated in a variety of tumors such as ovarian, breast, lung, colorectal, and prostate cancers, and hepatocellular carcinoma. For instance, overexpression of SPHK1 in a xenograft model of ovarian cancer enhanced tumor growth, followed by enhancement of proliferation and stemness [34]. Thus, SPHK1/S1P axis regulation became one of the main targets in cancer treatment. To date, a wide range of sphingosine, amidine, bicyclic aryl, amino-alcohol-based, and lipidic and non-lipidic small molecule SPHK inhibitors have been designed and applied in the areas of cancer, allergy and inflammation, transplantation, and viral infection treatment [35].

Dansylation of PF-543 structure was performed by labeling the core structure with 5-(dimethylamino)naphthalene-1-sulfonyl and, as dansyl group has fluorescence property, it was recommended to utilize it for fluorescent-based SPHK activity assays. SPHK1 inhibition assay showed that PF-543 and dansyl-PF-543 both have almost the same binding affinity and binding pattern as SPHK1 [33]. The (R)-2-(hydroxymethyl)-pyrrolidine group of PF-543 is terminal-1 bound substrate and replaces the position of a lipid head group. The pyrrolidine nitrogen and the hydroxyl of PF-543 form hydrogen bonds to the sidechain of Asp264 which is involved in sphingosine recognition. PF-543 was reported as a weak substrate for SPHK1. To phosphorylate the PF-543 structure, (R)-2-(hydroxymethyl)-pyrrolidine head group needs to be rotated for 180° to move the primary hydroxyl into a position of the lipid primary hydroxyl group. SPHK2 has three residue differences in the lipid binding site compared to SPHK1. Cys is in the position of Phe374, Val replaces Ile260 and Lue is Met358 in SPHK2. Those differences make the binding site larger in SPHK2. The phenyl ring of PF-543 binds against Phe374 and may be the tightest bound part of the molecule. Thus, Cys374 in SPHK2 may be a significant reason for the ∼132-fold selectivity of PF-543 for SPHK1 over SPHK2 [36].

In vitro application of PF-543 in some cells leads to autophagy and necrosis, but not apoptosis. For instance, the survival rate of Ca9-22 and HSC-3 cells diminished to 19.8% and 26.7%, respectively, in the presence of 25 μM PF-543. The morphology of the above-mentioned cells was also affected by PF-543 treatment [25]. In the case of HTC-116 cells, PF-543 time and dose dependently displayed anti-survival effect, which was revealed after 48 h incubation with 10 μM concentration of chemical [37]. It is known that ceramide triggers apoptosis through the stress-activated protein kinase (SAPK)/c-jun kinase (JNK) cascade [38]. Unlike PF-543, DPF-543 highly accumulates ceramide, and the impact of DPF-543 on the pro-apoptotic and anti-apoptotic signaling members of the BCL-2 family, including BAD and BCL-2, will be the next stage of our research.

Since the CCK-8 reagents have higher aqueous solubility than the other tetrazolium salts such as MTT, XTT, MTS, or WST, this cell viability kit was highly recommended for cell cytotoxicity, viability, and proliferation tests [39,40]. Although the dansylation on PF-543 provides considerable merits for the experimental visualization and simplicity for bioassay, the larger molecular weight and hydrophobicity in DPF-543 might be related to the enhancement of the membrane permeability and may allow it to stay longer in the cells to trigger cytotoxicity. Fortunately, the dansyl modification on PF-543 (DPF-543), far from the position of SPHK binding site, did not reduce its activity [33]. Previously, PF-543 was reported to not directly activate the CerSases, which is an essential enzyme to produce Cers [24]. The higher cytotoxicity of DPF-543 may come from the strong potency of the production of Cers. DPF-543 showed almost 5-times stronger cytotoxicity on LLC-PK1 cells than its prototype.

There were several examples of the regulation of Cer synthesis by SPHK inhibitors. N,N-dimethylsphingosine (DMS) induced apoptosis in hematopoietic and carcinoma-originated cancer cells by sphingosine 1-phosphate (S1P) reduction. This changed the ceramide/SPP rheostat in favor of cellular ceramide, which induced apoptotic progress in cancer cells [41,42]. The SPHK regulation has been investigated as a drug target for anticancer therapy by those motivated to design the selective SPHK inhibitors, F-12509, ABC294640, SKI-I, K145, and SKI-II [43,44,45,46,47,48]. The SPHK inhibitor in general not only reduced S1P but also simultaneously increased Cers. For example, the human gastric cancer HGC 27 cells treated with SKI II revealed reduced S1P levels and increased amounts of DHCers and Cers. The DHCer accumulation remained for 48 h, whereas Cers levels went to basal level within 24 h [48]. ABC294640, an SPHK2 selective inhibitor, downregulated the SK1 and SK2 mRNA expression in A498 kidney adenocarcinoma cells, and this initiated the decrease of S1P levels and increment of ceramide levels. Moreover, 48 h exposure of A498 cells to IC_50_ of ABC294640 also altered the expression and activation of signaling proteins, including STAT3, AKT, ERK, p21, p53, and FAK [44]. Another study showed that dose-dependent treatment of TRAMP-C2 cells by ABC294640 elicited active accumulation of dihydroceramides, suggesting that chemicals inhibit the dihydroceramide desaturase activity. TRAMP-C2-cells-injected xenograft model mice also displayed similar results. Immunoblotting results revealed that ABC294640 only downregulates the DEGS activity but not the expression level [45]. Recently, some anticancer drugs showed non-targeted impact on SPHK/S1P/S1PR signaling pathway. For instance, doxorubicin and etoposide treatment of parental HL-60 cells leads to significant abatement in SphK1 activity within 30 min of exposure and reaches to a peak at 90–120 min. In accordance with the SphK1 downregulation, both chemicals strongly activated accumulation of the proapoptotic lipid ceramide, peaking after 90–120 min of contact [43]. F-12509 is a sesquiterpene quinone derived from *Trichopezizella barbata*, which acts as a competitive inhibitor for SphK1, strongly inhibiting the SPHK1 activity in chemoresistant HL-60/Doxo cells or HL-60/VP16 cells. Interestingly, increase of ceramide synthesis was contemporaneously linked with suppression of SphK1 activity [43]. Plant-derived compounds such as cannabinoids may also indirectly influence sphingolipid metabolism. For example, G-coupled protein receptor cannabinoid1 (CB1) activation increased ceramide levels in primary astrocytes and glioma cells via factor associated with neutral sphingomyelinase activation which induces the breakdown of sphingomyelin into ceramide and phosphorylcholine [49]. Delta9-tetrahydrocannabinol and the synthetic cannabinoid agonist WIN-55,212-2 administration significantly regressed the malignant gliomas in Wistar rats and in mice with recombination-activating gene 2 deficiency. Results revealed that cannabinoids induce apoptosis signal via cannabinoid receptors, ceramide accumulation, and Raf1/extracellular-signal-regulated kinase activation in two subclones of C6 glioma cells [50]. Paclitaxel is a member of the taxane family, with diterpen structure, actively used in chemotherapy for different types of cancer [51]. Ceramide yield was increased 2- and 2.5-fold when hormone-independent MDA-MB-468 and hormone-dependent MCF-7 breast cancer cells were exposed to taxol in doses of 50 nM and 1.0 µM, respectively [52]. The combination of paclitaxel with C6-ceramide in biodegradable poly(ethylene oxide)-modified poly(epsilon-caprolactone (PEO-PCL) nanoparticles significantly overcame the drug resistance, enhanced the tumor growth delay, and increased tumor volume doubling time in human adenocarcinoma xenografts [53].

In our results, the enhancement of hydrophobicity by dansylation on PF-543 gave a great enhancement of CerSases activities, resulting in an endogenous Cers increase via de novo sphingolipid pathway (Figure 6a). In this CerSase assay, the Cers conversion from DHCers by DHCer desaturase inserting a double bond on C4-C5 position of sphingoid bases was shown to be an inefficient conversion ratio. Most of the conversion ratio of Cers from DHCers was far below 100%, indicating the desaturation process is inefficient and thus is a rate-limiting step to produce the Cers (Appendix A). Nevertheless, the conversion ratio on long-chained Cers of C22:0 and C24:1 was relatively higher in DPF-543 treatment, which raised the DHCers by CerSases activation. Compared to the Cers increase by PF-543, DPF-543 selectively causes at least a two-fold Cers increase of C22:0 and C24:1, which might reflect a stronger cytotoxicity of DPF-543.

There is evidence that lysosomal aSMase plays an important role in ceramide formation after stimulation with chemotherapeutic agents, luteolin or fluphenazine [54,55]. Here, although the aSMase activation is not so strong, PF-543 and DPF-543 also triggered the lysosomal aSMase activation, indicating increased cytotoxicity via pro-apoptotic Cers accumulation by sphingomyelin (SM) hydrolysis (Figure 5). In the same conditions, DPF-543 induced almost 4-fold Cers accumulation where PF-543 just showed a 1.5-fold Cers increase, suggesting that the Cers increase by DPF-543 might be a combined pool not only from aSMase activation but also from additional de novo synthetic pathway (Figure 6a). Most of the endogenous Cers was highly accumulated by DPF-543, which increased toxicity coming from dansylation of PF-543 and increased hydrophobicity and both activations of aSMase and CerSases in de novo pathway (Figure 6b).

Compared to PF-543 effects, DPF-543 preferentially activates de novo pathway in total Cers accumulation. We further investigated whether DPF-543 could increase the Cers amounts during co-treatment of SPT inhibitor Myr or CerSs inhibitor FB1. As expected, the pretreatment of 10 μM Myr, which is a high concentration to block the SPT activity, reduced the total Cers amount which may be synthesized from de novo pathway. PF-543 likely acts to restore Cers concentration to control level (Figure 7a), which was similarly observed when FB1 application reduced Cers levels (Figure 8a). The DPF-543 effect with Myr or FB1 pretreatment on the total Cers was evidently stronger than the case of PF-543. The reason for unexpected Cers increase by DPF-543 may be explained as follows: DPF-543 stays longer in hydrophobic cellular organs, including the ER membrane where de novo sphingolipid biosynthesis started with SPT activation. Another hypothesis on the S1P reduction by SPHK inhibition by DPF-543 is that it caused a sharp change in Cer/S1P rheostat balance.

Taken together, DPF-543 treatment strongly activated a series of enzymes located in the de novo sphingolipid pathway (Figure 10). Although these effects on Cers accumulation are not fully explained yet, the pharmacological ability for SPA accumulation by different SPHK inhibitors is a key component to explain their different cytotoxicity. In our study, the structural design of dansylation of PF-543 with a higher hydrophobicity provided the stronger cytotoxicity caused by Cers accumulation, mainly by SPA-induced CerSs activation.

## 4. Materials and Methods

### 4.1. Materials

Ceramide standards with different carbonyl chains of 16, 17, 18:1, 18, 24, and 24:1 were obtained from Matreya LLC (State College, PA, USA). C17-sphinganine (C17-Sa) and fumonisin B1 (FB1) were purchased from Cayman Chemical (Ann Arbor, MI, USA). Myriocin and other reagents for SPT assay were from Sigma Aldrich (St. Louis, MO, USA). PF-543 was obtained from Echelon biosciences (Salt Lake City, UT, USA). DPF-543 was synthesized from Mokpo University and evaluated as a SPHK inhibitor [33]. Other organic solvents were purchased from Honeywell Burdick & Jackson (Charlotte, NC, USA). The reagents for cell culture media were purchased from HyClone (South Logan, UT, USA). Primary antibodies of serine palmitoyltransferase long chain base subunit 1 (Sptlc1), ceramide synthase 4 (CerS4), CerS6, and horseradish peroxidase (HRP) labeled secondary antibodies were from Thermo (Waltham, MA, USA). The β-actin antibody was from Merk Millipore (Burlington, MA, USA). Sphingomyelinase (SMase) activity kit was from Abcam (Cambridge, MA, USA). Fatty-acid-free BSA, BSA fraction V, and protease inhibitor cocktail were purchased from Roche (Mannheim, Germany). The isotope-labeled L-serine (3,3-d2) was purchased from Cambridge Isotope Laboratories, Inc. (Tewksbury, MA, USA).

### 4.2. Cell Culture

A porcine kidney proximal tubule cell line, LLC-PK1 cells, was cultured in Dulbecco’s Modified Eagle’s medium. A human kidney proximal tubular cell, HK-2 cells, was in RPMI medium. Each cell was subcultured and was supplemented with 10% fetal bovine serum (FBS) and antibiotics of 100 units/mL of penicillin, and 100 μg/mL of streptomycin (P/S). The cells were grown at 37 °C in a 5% CO_2_.

### 4.3. Cell Viability Test

Ten thousand LLC-PK1 cells were seeded onto a 96-well plate and cultured for 24 h in 200 µL of DMEM with 10% FBS plus P/S. Standard stock solutions of PF-543 and DPF-543 (40 mM in DMSO) were diluted serially with DMSO in the range of 1000 µM to 1.95 µM. Grown cells were each treated with 1 µL of PF-543 or DPF-543 solutions after changing the old media for fresh ones (100 µL). After cells were incubated for 24 h, cultured media were removed by gentle aspiration and replaced with 100 µL of PBS. Then, 10 µL of cell counting kit-8 (CCK-8) solution (Dojindo Molecular Technologies Inc., Rockville, MD, USA) was spiked into each well. Plates were incubated for 3 h at 37 °C to complete the orange color of formazan production on living cells. Plates were read with a microplate reader, absorbance at 450 nm.

### 4.4. Regulation of Sphingolipid Metabolism

To investigate PF-543 and DPF-543 effects on SPT activity, 10 µM of myriocin was spiked to LLC-PK1 cells and was incubated for 24 h before PF and DPF-543 treatment. LLC-PK1 cells were treated with either 20 µM PF-543 or DPF-543. In control, the same volume of DMSO was spiked to the cells. Cells were incubated for an additional 24 h before harvest.

To study PF-543 and DPF-543 effects on CerSs activity, 35 µM FB1 was spiked to the cells and incubated following the same protocol as noted above.

### 4.5. Lipid Extraction

Under the 200 µL of RIPA lysis buffer with complete protease inhibitor cocktail, harvested cell pellets were homogenized. Protein amount was determined according to the manufacturer’s instructions of the Thermo protein assay kit (Pierce, IL, USA). Five hundred pmol of internal standard C18:1/C17 ceramide and 750 µL of MeOH:CHCl_3_ (2:1, *v/v*) were added into 100 µg of protein lysate. The mixture was incubated overnight at 48 °C. After cooling down to ambient temperature, 75 µL of 1M KOH in MeOH was used for complete lipid digestion for 2 h at 37 °C with vigorous shaking. We spiked acetic acid for mixture neutralization. The mixture (750 µL) was transferred into new tubes. For lipid extraction, 350 µL of CHCl_3_ and 150 µL of distilled water were added and vortexed, centrifuged for 5 min at 14,000 rpm. Aqueous phase was re-extracted with additional CHCl_3._ Combined lower organic phases were dried. Dried residues were reconstituted with MeOH to introduce mass spectrometry.

### 4.6. SPT Activity

SPT activity assay was carried out in LLC-PK1 cells as reported previously with partial modification. The assay mixture composition was a solution of 100 mM HEPES, pH 8, 5 mM DTT, 50 µM PLP, 100 µM Pal-CoA, 10 mM 3,3-D2-serine, and the complete protease inhibitor cocktail [56]. The SPT assay started by adding cell lysates of 200 µg of protein into 200 µL of assay mixture, incubated at 37 °C for 30 min. After the stop solution (CHCl 3: MeOH (2:1); 500 µL) was added, 50 pmol of C17-sphingosine was spiked as an internal standard for chromatographic separation and quantitation. The tubes were vortexed vigorously for 5 min and centrifuged for 5 min at 14,000 rpm. Lower organic phase with an additional 200 µL of CHCl_3_ was combined and dried.

### 4.7. CerS Activity

The assay mixture contained 50 mM HEPES-KOH, pH 7.4, 25 mM KCl, 2 mM MgCl_2_, 0.5 mM DTT, 0.1% fatty-acid-free BSA, 50 µM Pal-CoA, and 10 µM C17-sphinganine [57]. The cell lysate of 100 µg of protein was incubated at 37 °C for 1 h with 200 µL of assay mixture. Reaction was terminated by 500 µL of CHCl_3_:MeOH (2:1; *v/v*) stop solution. The tube containing newly synthesized C17-based ceramide was vortexed and centrifuged for 5 min at 14,000 rpm. Lower organic phase with an additional 200 µL of CHCl_3_ extract was combined and dried.

### 4.8. aSMase Activity

The aSMase activity assay was measured by fluorometric SMase assay method (Abcam) following the manufacturer’s protocol. The LLC-PK1 cells were incubated with 20 µM of PF-543 and DPF-543 for 24 h. The supernatant from the cell lysate was transferred into a new tube. The final volume, 50 µL containing 50 µg protein, was loaded into 96-well plate. Then, 50 µL of sphingomyelin (SM) solution was applied, and the plate was incubated at 37 °C for 3 h. After that, the 50 µL of red indicator for SMase product was spiked to each well. The plate protected from light was kept at ambient temperature for 2 h. Fluorescence intensity was measured on a microplate reader at Ex/Em 485/535 nm. The significance of relative fluorescence unit (RFU) values was calculated by ANOVA.

### 4.9. Immunoblotting

For immunoblotting to the expression of sphingolipid metabolic enzymes, human proximal tubule epithelial HK-2 cells were used. The collected HK-2 cells were washed twice in ice-cold PBS solution and lysed in 200 µL of RIPA lysis buffer (25 mM Tris•HCl pH 7.6, 150 mM NaCl, 1% NP-40, 1% sodium deoxycholate, 0.1% SDS). The BCA protein assay kit (Thermo, Pierce, IL, USA) was used to determine protein concentration. The loaded samples containing 20 µg/10 µL protein concentration were separated on a 10% SDS-PAGE gel and transferred to PVDF membranes by semi-dry method. Then, blotting status was checked by using Ponceau S reagent, and membranes were washed with TBST three times for 10 min. The membranes were blocked in 5% BSA in TBST overnight and then incubated with the primary anti-Sptlc1, CerS4, and CerS6 antibodies (1:1000 dilution in 5% BSA-TBST) overnight at 4 °C. Anti β-actin clone c4 mouse monoclonal antibody was prepared in 1:10,000 dilution. After incubation, the membranes were washed with TBST three times for 10 min. The blots were then incubated overnight with secondary horseradish-peroxidase-conjugated goat anti-Rabbit IgG antibody (1:5000 dilution in 5% BSA in TBST) and goat anti-Mouse IgG (H+L) antibody (1:25,000 dilution in 5% BSA in TBST) to detect targeted proteins and β-actin, respectively. The membranes were then washed three times with TBST. The protein bands were visualized by Pierce ECL Plus Western Blotting Substrate and read in Amersham Imager 600 (GE Healthcare Bio-Sciences AB, Sweden). Data were represented in Appendix A.

### 4.10. LC-MS/MS Conditions

AB Sciex QTRAP 4500 model mass spectrometry coupled with Shimadzu UPLC system was applied to determine the amounts of ceramides. Mass spectrometry was operated in positive ion mode. Precursor and product ions were displayed in Appendix A with optimal parameters. Linear calibration curves were constructed by using the area ratio of standard ceramides to IS C17-ceramide to build calibration curves. All Cers were well separated on Shiseido Capcell Pak C18 MG III type, 50 × 3 mm, 5 µM. Mobile phase A was 10 mM ammonium acetate in water with 0.1% formic acid and mobile phase B was 10 mM ammonium acetate in acetonitrile:2-propanol (4:3; *v/v*) with 0.1% formic acid. Gradient elution with a flow rate of 0.3 mL/min started at 85% of B, increased to 100% B after 1.5 min and was held for 10 min, then was re-equilibrated to 85% B for 5 min. The mass spectrometry parameters for C17-Cers and C17-DHCers and other sphingolipid metabolites are listed in Appendix A, respectively.

## Figures and Tables

**Figure 1 ijms-22-09190-f001:**
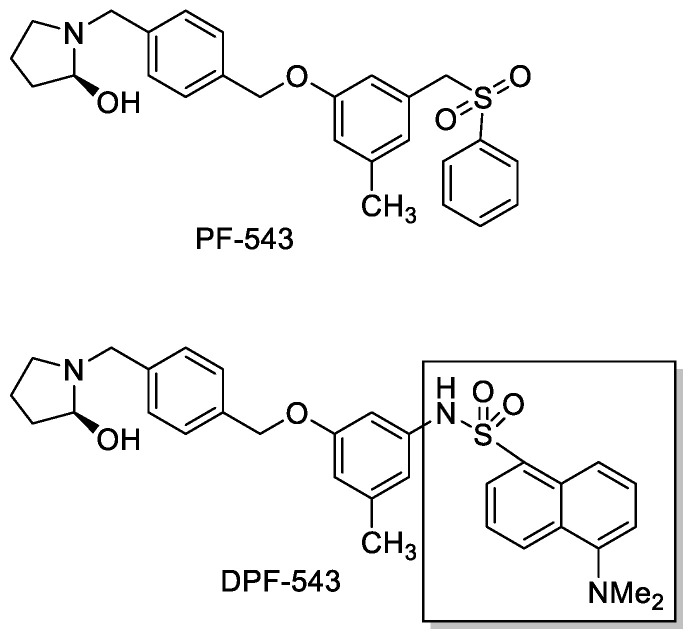
Structure of PF-543 and DPF-543.

**Figure 2 ijms-22-09190-f002:**
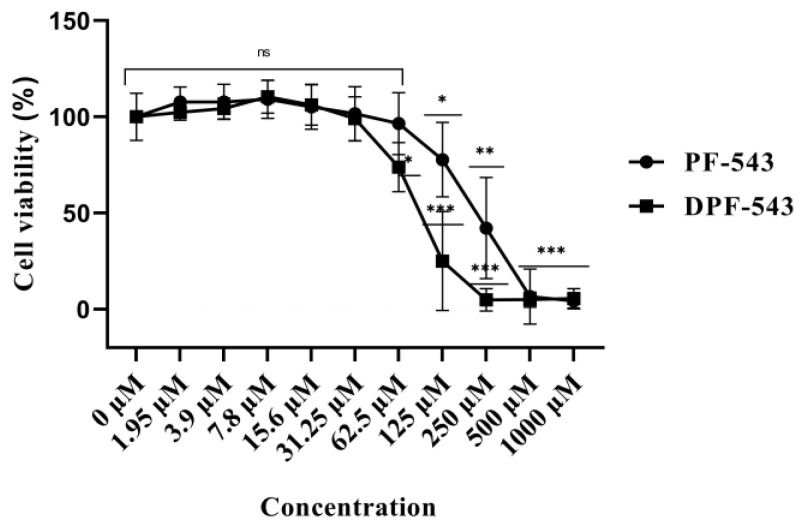
Cell viability of PF-543 and DPF-543 in LLC-PK1 cells. Statistical significance of the data was calculated based on the difference between chemical applied groups and 0 µM treated cells. Data are shown as means ± RSD, ns-not significant, * *p* < 0.05, ** *p* < 0.01, *** *p* < 0.001.

**Figure 3 ijms-22-09190-f003:**
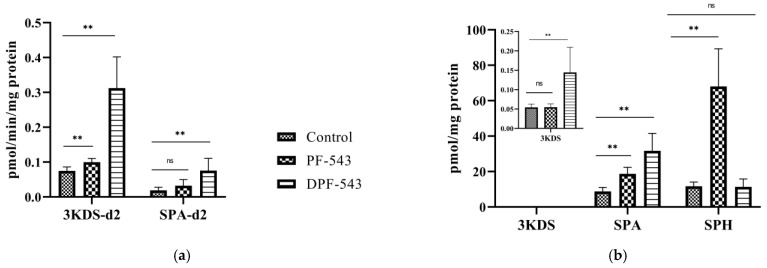
(**a**) Deutralized KDS and SPA accumulation by PF-543 and DPF-543; (**b**) PF-543 and DPF-543 impact on endogenous sphingoid bases. Data are shown as means ± SD, ns—not significant, ** *p* < 0.01.

**Figure 4 ijms-22-09190-f004:**
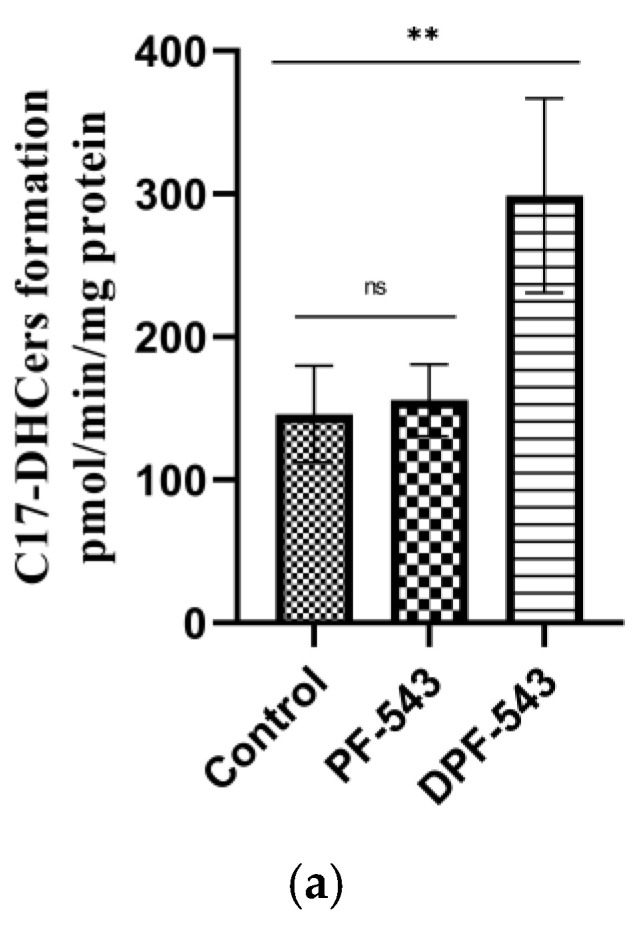
(**a**) Total C17-DHCer amount in PF-543 and DPF-543 application; (**b**) individual C17-DHCer accumulation by PF-543 and DPF-543 treatment; (**c**) total C17-Cer amount in PF-543 and DPF-543 application; (**d**) individual C17-Cer alteration by PF-543 and DPF-543. Data are shown as means ± SD, ns—not significant, * *p* < 0.05, ** *p* < 0.01, *** *p* < 0.001.

**Figure 5 ijms-22-09190-f005:**
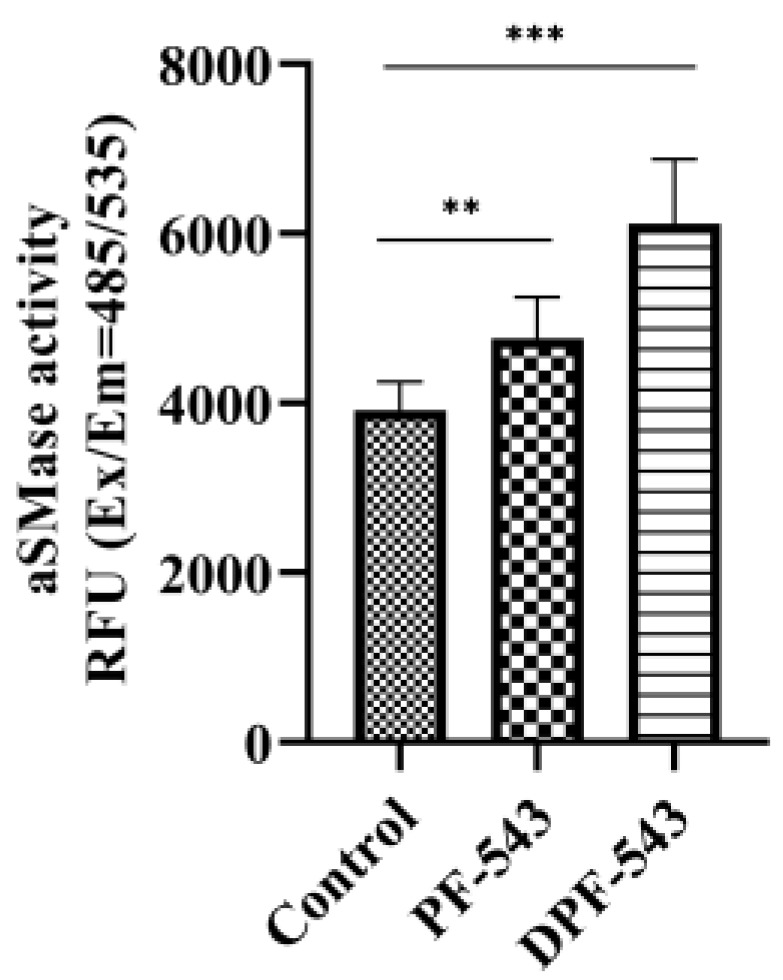
Acid sphingomyelinase activation by PF-543 and DPF-543 treatment. Data are shown as means ± SD, ns—not significant, ** *p* < 0.01, *** *p* < 0.001.

**Figure 6 ijms-22-09190-f006:**
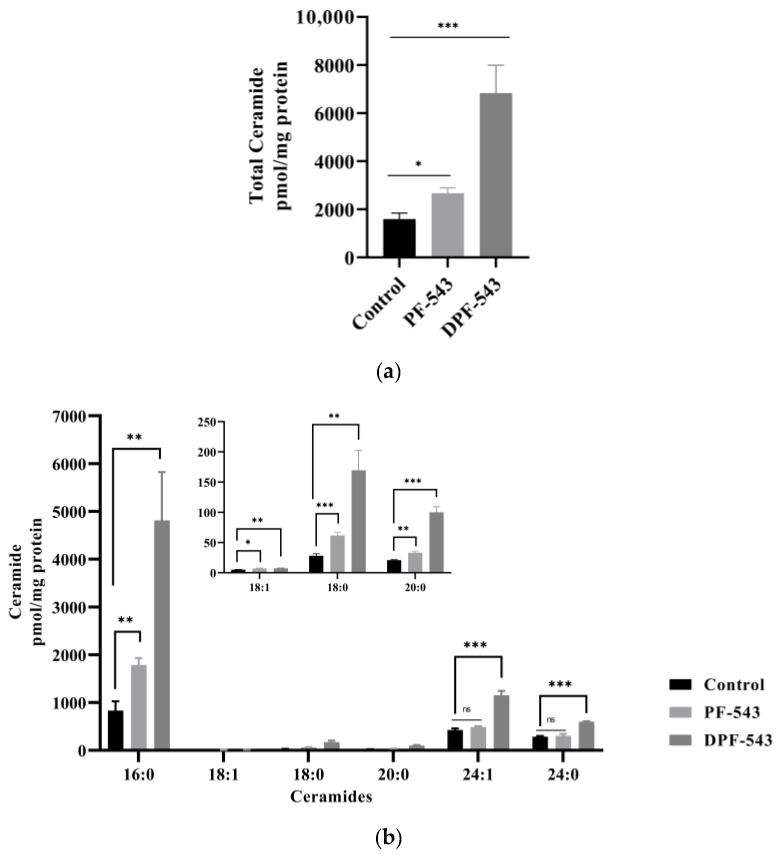
(**a**) Total endogenous ceramide accumulation by SPHK1 inhibitors; (**b**) alteration of endogenous ceramides by SPHK1 inhibitors. Data are shown as means ± SD, ns—not significant, * *p* < 0.05, ** *p* < 0.01, *** *p* < 0.001.

**Figure 7 ijms-22-09190-f007:**
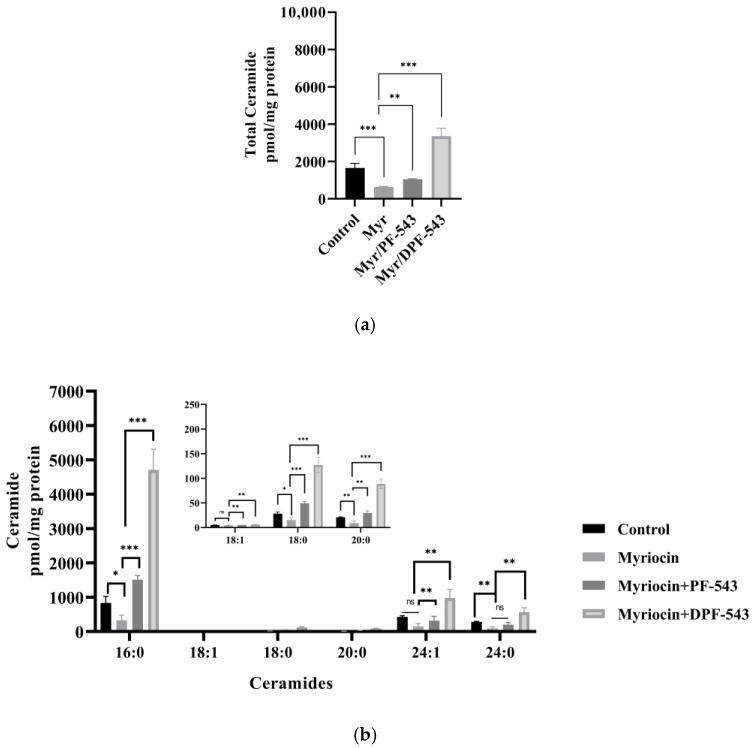
(**a**) PF-543 and DPF-543 impact on total ceramide accumulation reduced by myriocin treatment; (**b**) alteration of individual ceramides by PF-543 and DPF-543 after myriocin treatment. Data are shown as means ± SD, ns—not significant, * *p* < 0.05, ** *p* < 0.01, *** *p* < 0.001.

**Figure 8 ijms-22-09190-f008:**
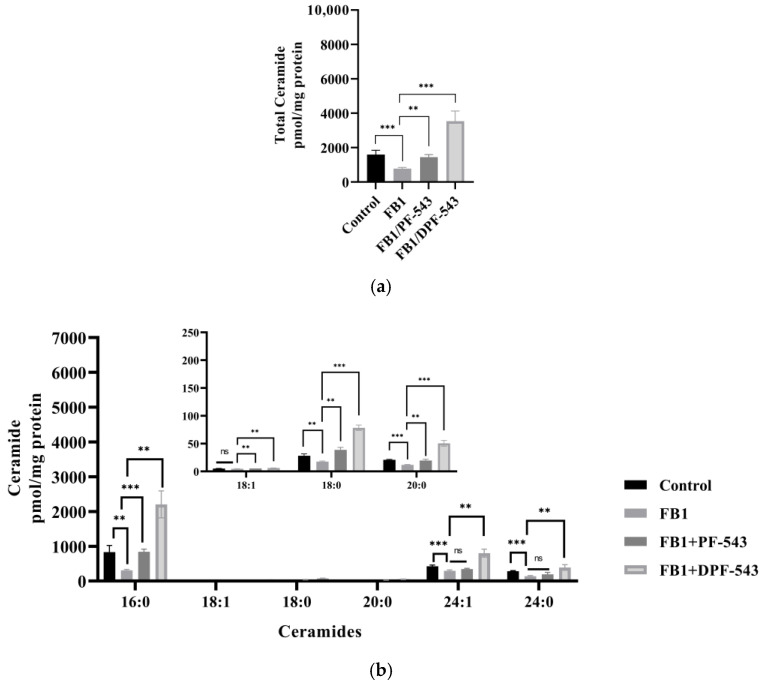
(**a**) PF-543 and DPF-543 impact on total ceramide accumulation reduced by FB-1 treatment; (**b**) alteration of individual ceramides by PF-543 and DPF-543 after FB-1 treatment. Data are shown as means ± SD, ns—not significant, ** *p* < 0.01, *** *p* < 0.001.

**Figure 9 ijms-22-09190-f009:**
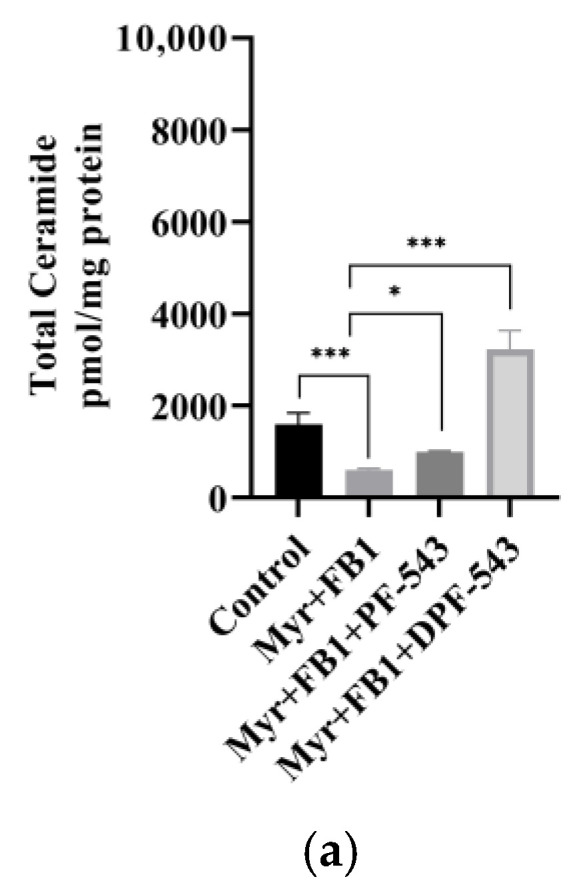
(**a**) PF-543 and DPF-543 impact on total ceramide accumulation mitigated by combined treatment of myriocin with FB-1; (**b**) alteration of individual ceramides by PF-543 and DPF-543 after combined treatment of myriocin with FB-1. Data are shown as means ± SD, ns—not significant, * *p* < 0.05, ** *p* < 0.01, *** *p* < 0.001.

**Figure 10 ijms-22-09190-f010:**
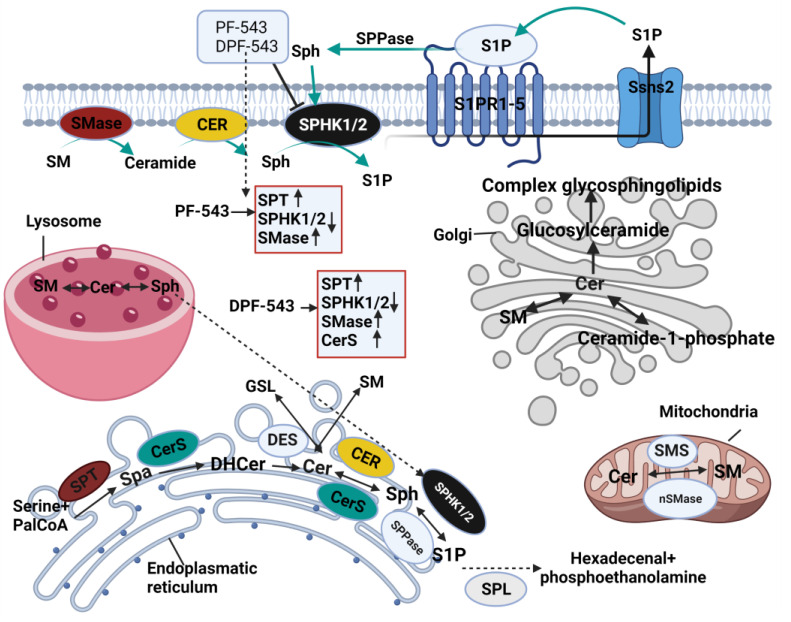
Action mechanism of PF-543 and DPF-543 on cellular sphingolipid metabolism. Figure was created by BioRender.com, accessed on 3 August 2021. Enzyme colors were defined based on inhibitor action: If the both chemical inhibits: Black; If the both inhibitor activate: Burgundy; If PF543 only activates: Yellow; If DPF543 only activates: Sky blue.

## Data Availability

Not applicable.

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
