# Peer review of "A Dansyl-Modified Sphingosine Kinase Inhibitor DPF-543 Enhanced De Novo Ceramide Generation"

_ijms, 2021, doi:10.3390/ijms22179190_

Round 1

Reviewer 1 Report

In the present paper, M. Shamshiddinova and coworkers investigated the comparative efficiency of dansyl-modified sphingosine kinase inhibitor PF-543 (DPF-543) on the Cers synthesis along with PF-543. The authors concluded that DPF-543 relatively enhanced Cers accumulation via de novo pathway which was not observed in PF-543. Moreover, the present results demonstrated that the structural modification on sphingosine kinase inhibitors is still an attractive anti-cancer strategy by regulating sphingolipid metabolism. Overall, I think that the paper could be of interest for readers of to the readers of "International Journal of Molecular Sciences" and researchers, in general.

I would like to make some suggestions on how to make the paper stronger.

Please review the recent literature and comparing interesting data obtained with dansyl-modified DPF-543 with results obtained by cannabinoids (for your convenience, you could consider a recent paper of Pagano et al., Int. J. Mol. Sci. 2021, 22, 3680).

Velasco et al. in 2005 (Life sciences, vol. 77, 14, pages 1723-1731) wrote a very interesting manuscript “Cannabinoids and ceramide: Two lipids acting hand-by-hand”. So, based on the results observed in this study, what is justified to expect in terms of physiological and therapeutic effects?

Based on these data, it is possible predict if DPF-543 can cause side effects in vivo?

Did you explore (or you plan to explore) the role of other signaling pathway directly/not directly related to sphingolipid pathway activation? Please add two/three sentences in discussion section of revised manuscript.

In light of the results here obtained, please to discuss on the possible application of nutraceutics and/or antioxidants/antinflammatory/anticancerogens compounds that, acting on molecular signaling pathway explored in the present paper, could provide a possible strategy to prevent and counteract cancers in humans.

The authors could add a Graphical abstract  showing the molecular signaling pathways directly/not directly explored in these experiments; in this way, I feel that the readers can better understand the pathophysiological cross-talk studied in the present paper, adding new evidence on the potential therapeutic effects of sphingosine kinase inhibitors and specifically of DPF-543.

Author Response

REV 1

  • Please review the recent literature and comparing interesting data obtained with dansyl-modified DPF-543 with results obtained by cannabinoids
  • In discussion section p14, lane 358-368 we inserted the additional information about how cannabinoids induce neutral sphingomyelinase activity and increased the ceramide level in primary astrocytes and glioma cells. Moreover, the data about natural cannabinoid and synthetic cannabinoid agonist WIN55,212-2 role in regression of malignant gliomas in rodents were also included in this section.
  • We inserted this in discussion section that “Plant derived compounds such as cannabinoids also indirectly may influence on sphingolipid metabolism. For example, G-coupled protein receptor CB1 (Cannabinoid1) activation increased ceramide levels in primary astrocytes and glioma cells via factor associated with neutral sphingomyelinase activation which induces the breakdown of sphingomyelin into ceramide and phosphorylcholine. Delta9-tetrahydrocannabinol and the synthetic cannabinoid agonist WIN-55,212-2 administration significantly regressed the malignant gliomas in Wistar rats and in mice with recombination activating gene 2 deficiency. Results revealed that cannabinoids induce apoptosis signal via cannabinoid receptors, ceramide accumulation and Raf1/extracellular signal-regulated kinase activation in two subclones of C6 glioma cells.”

  • Based on these data, it is possible predict if DPF-543 can cause side effects in vivo?
  • There is not in vivo data about the side effects of PF-543. According to the results of conducted research we may postulate that unlike PF-543, dansylation of the core structure increased the ceramide accumulation, and therefore cell viability was decreased significantly. As we know from previously published papers, ceramide accumulation causes the cell death and apoptosis, and we hypothesize that DPF-543 may act stronger that PF-543 in cancer treatment. We still have not conducted in vivo experiments on searching side effects of DPF-543, this will be the next step in our research.

  • Did you explore (or you plan to explore) the role of other signaling pathway directly/not directly related to sphingolipid pathway activation? Please add two/three sentences in discussion section of revised manuscript.
  • On page 13, lane 311-316 you will find the text about the plan on exploration of DPF-543 on the other signaling pathway which may directly or indirectly linked with sphingolipid signaling route.
  • We inserted the following sentence in discussion section that “It is known that ceramide trigger the apoptosis through the stress-activated protein kinase (SAPK)/c-Jun kinase (JNK) cascade[38]. Unlike PF-543, DPF-543 highly accumulates the ceramide, and the impact of DPF-543 on the pro-apoptotic and anti-apoptotic signaling members of the BCL-2 family, including BAD and BCL-2 will be the next stage of our research.”

  • In light of the results here obtained, please to discuss on the possible application of nutraceutics and/or antioxidants/antinflammatory/anticancerogens compounds that, acting on molecular signaling pathway explored in the present paper, could provide a possible strategy to prevent and counteract cancers in humans.
  • On the pages 13-14, lane 339-375 you will find the information about natural or synthetic chemicals which are widely applied in cancer treatment. In that section, we described how they directly or indirectly influence on enzyme activities which is actively involved in sphingolipid metabolism.
  • We inserted that “ABC294640 is a SPHK2 selective inhibitor downregulated the SK1 and SK2 mRNA expression in A498 kidney adenocarcinoma cells, by this initiated the decrease of S1P levels and increment of ceramide levels. Moreover, 48 hour exposure of A498 cells to IC50 of ABC294640 also altered the expression and activation of signaling proteins, including STAT3, AKT, ERK, p21, p53 and FAK[44]. Another study showed that, dose dependent treatment of TRAMP-C2 cells by ABC294640 elicited active accumulation of dihydroceramides suggesting that chemical inhibits the dihydroceramide desaturase activity. TRAMP-C2 cells injected Xenograft model mice also displayed similar results. Immunoblotting results revealed that, ABC294640 only downregulates the DEGS activity but not expression level. Recently, some anticancer drugs showed non-targeted impact on SPHK/S1P/S1PR signaling pathway. For instance, Doxorubicin and etoposide treatment of parental HL-60 cells lead to significant abatement in SphK1 activity within 30 min of exposure and reaching to a peak at 90–120 min. In accordance with the SphK1 downregulation, both chemicals strongly activated accumulation of the proapoptotic lipid ceramide peaking after 90–120 min of contact. F-12509 is a sesquiterpene quinone derived from a Trichopezizella discomycete, acts as a competitive inhibitor for SphK1, strongly inhibited the SPHK1 activity in chemoresistant HL-60/Doxo cells or HL-60/VP16 cells. Interestingly, increase of ceramide synthesis was contemporaneously linked with suppression of SphK1 activity. Plant derived compounds such as cannabinoids also indirectly may influence on sphingolipid metabolism. For example, G-coupled protein receptor CB1 (Cannabinoid1) activation increased ceramide levels in primary astrocytes and glioma cells via factor associated with neutral sphingomyelinase activation which induces the breakdown of sphingomyelin into ceramide and phosphorylcholine. Delta9-tetrahydrocannabinol and the synthetic cannabinoid agonist WIN-55,212-2 administration significantly regressed the malignant gliomas in Wistar rats and in mice with recombination activating gene 2 deficiency. Results revealed that cannabinoids induce apoptosis signal via cannabinoid receptors, ceramide accumulation and Raf1/extracellular signal-regulated kinase activation in two subclones of C6 glioma cells. Paclitaxel member of taxanes with diterpene structure actively used in chemotherapy of different types of cancer. Ceramide yield was increased 2- and 2.5-folds when hormone-independent MDA-MB-468 and hormone-dependent MCF-7 breast cancer cells were exposed to Taxol in a dose of 50nM and 1.0µM, respectively. Combination of paclitaxel with C6-ceramide in biodegradable poly (ethylene oxide)-modified poly (epsilon-caprolactone (PEO-PCL) nanoparticles significantly overcome the drug-resistance, enhanced the tumor growth delay and increased tumor volume doubling time in human adenocarcinoma xenografts.”

  • The authors could add a Graphical abstract showing the molecular signaling pathways directly/not directly explored in these experiments; in this way, I feel that the readers can better understand the pathophysiological cross-talk studied in the present paper, adding new evidence on the potential therapeutic effects of sphingosine kinase inhibitors and specifically of DPF-543.
  • PF-543 and DPF-543 impact on cellular sphingolipid metabolic enzymes was displayed in Fig. 10 on p 16, between the lanes 418 and 419. In the graph you can find cellular sphingolipid metabolism and the enzymes which are involved in this signaling pathway. Enzymes which were activated and suppressed by inhibitors were indicated with arrows.

Reviewer 2 Report

Introduction

I suggest to improve Introduction section. There is no aim as well there is no mention about pharmacological or preventive effects of PF-543 (DPF-543). Please revise it.

Results and Figures

General comment: Please revise all statistical analysis and graphs. There is not appropriate to mention any results without to mention statistical significance in the main text as well in Figure legend. As well, wherever the authors mentioned * symbol, I suggest to specify what that means, always.

Point 1. Please change heading of 2.1 result “Cell viability evaluation following DPF-543 treatment” instead of the old one. In Figure 2 as well in 2.1 Results, the authors mentioned “two chemicals gave the irreversible toxicity with 5% cell viability… ”, thus it is appropriate to add in cell viability graph a symbol that represents statistical significance compared to control cell (or 0 mM treated cells). I suggest to more explanation about it in Figure 2 caption.

Point 2. Please add in Figure 4d, 6a, 7a and 9a (graphs and respective legend), some information about statistical significance as well some symbols for better understanding those results (if not, please revise saying “no significant changes were observed…”).

Please take care about abbreviations throughout the main text (USA instead of US and etc.)

Author Response

REV2

  • I suggest to improve Introduction section. There is no aim as well there is no mention about pharmacological or preventive effects of PF-543 (DPF-543). Please revise it.
  • Introduction part p3, lane 77-98 you will find pharmacological activities of PF-543. There is no report about pharmacological study on DPF-543 as it’s a newly synthesized compound. However, you may find the results of the docking study in the lane 104-108. As an aim of research, we added the reason about the need of this experimental study on the lane 109-111.
  • We inserted that “PF-543 decreased the expression of profibrotic markers such as mtDNA damage and fibrogenic monocyte recruitment in mice lungs with pulmonary fibrosis induced by bleomycin- and asbestos. Also, post-treatment of lung epithelial cells with PF-543 suppressed pulmonary fibrosis at the expense of reduced lung mtDNA damage and monocyte recruitment. Furthermore, SPHK1 inhibition by PF-543 impaired YAP1 co-localization with FSP1 in mice lung fibroblasts. In vitro studies revealed that, PF-543 treatment reduced the TGF-β- or BLM-induced mitochondrial reactive oxygen species (mtROS) in human lung fibroblasts (HLFs) and the expression of fibronectin (FN) and alpha-smooth muscle actin (α-SMA). These results suggest that PF-543 attenuated the TGF-β-induced YAP1 activation and mtROS generation, causing fibroblast activation, a vital inducer of pulmonary fibrosis. SPHK1 inhibition by PF-543 decreased matrix mineralization, alkaline phosphatase activity and the mRNA expression of Runx2 and Bglap in chondrocytes and osteoblasts, by this making it one of the promising candidates for spondylarthritis treatment. Intraperitoneal administration of PF543 improved endothelial function of arteries of hypertensive mice by decreasing endothelial nitric oxide synthase phosphorylation. Importantly, pharmacological inhibition of SPHK1 by PF-543 also reduced cardiac hypertrophy and endothelial dysfunction which were induced by Ang II. Other study showed that, PF-543, as a specific inhibitor of SphK1, could partially minimize the detrimental effects on lung injury of cecal ligation and puncture mice. PF-543 suppressed the SPHK1/S1P axis and by this mitigated the lung injury caused by sepsis in acute ethanol intoxication in rats.”
  • We also added “Novel PF-543 derivatives were evaluated on the activity of SPHK1/2 inhibition. However, there is lack of data about their impact on other enzymes which actively take part on sphingolipid metabolism.”

Results and Figures

General comment: Please revise all statistical analysis and graphs. There is not appropriate to mention any results without to mention statistical significance in the main text as well in Figure legend. As well, wherever the authors mentioned * symbol, I suggest to specify what that means, always.

  • All the figures revised, and all statistically significant data were labeled with *, **, *** and ns according to their p value

Point 1. Please change heading of 2.1 result “Cell viability evaluation following DPF-543 treatment” instead of the old one. In Figure 2 as well in 2.1 Results, the authors mentioned “two chemicals gave the irreversible toxicity with 5% cell viability… ”, thus it is appropriate to add in cell viability graph a symbol that represents statistical significance compared to control cell (or 0 mM treated cells). I suggest to more explanation about it in Figure 2 caption.

  • The heading of the 2.1 result on page 4, lane 122 was changed into “Cell viability evaluation following DPF-543 treatment”. Also, the symbols that represents statistical significance of the data was added into graph.

Point 2. Please add in Figure 4d, 6a, 7a and 9a (graphs and respective legend), some information about statistical significance as well some symbols for better understanding those results (if not, please revise saying “no significant changes were observed…”).

  • All the figures listed above was revised and all data was labeled with ns, *, **, *** according to their statistical significance.

Please take care about abbreviations throughout the main text (USA instead of US and etc.)--- All abbreviations were checked and USA in the main text was changed with US.

Round 2

Reviewer 1 Report

The authors have satisfactorily responded to all my questions and made the necessary changes to the manuscript.

Reviewer 2 Report

Dear Editor

I accept revised form